# EFFECT OF LOCAL OSCILLATIONS ON THE SCALING LAWS OF DEEP NEURAL NETWORKS

## ABSTRACT

Deep neural network (DNN) scaling laws characterize how a model's performance (e.g. test loss) improves as a function of resources such as training data size, model parameters, or compute. These laws hold for a wide variety of model and data types. Empirical and theoretical results have found that the parameters of the scaling laws depend on aspects of the target data function such as continuity class and dimension. Here we show that another feature of the data, namely the local oscillatory complexity (LOC) of the target function, can dramatically alter scaling behavior. In particular, when the target function is highly oscillatory (parity-like), the drop in loss with more training data becomes shallower. We formalize a metric for local oscillatory complexity and study a family of parity-like target functions where this complexity is controlled by a frequency parameter. We show that high oscillatory complexity can shift the scaling curve upward (higher error floor), change the scaling exponent, and induce an earlier saturation regime. In our experiments, DNNs fail to benefit from additional data when the target function is highly oscillatory. These findings reveal that data continuity class and dimension are insufficient to guarantee standard scaling behavior – LOC must also be accounted for.

## 1 INTRODUCTION

Deep neural network (DNN) scaling laws – empirical power-law relationships predicting how model performance scales with data, model size, or compute – have become a cornerstone for designing modern deep learning models. These laws have been validated across many architectures (feedforward neural networks (FNNs), recurrent neural networks (RNNs), convolutional neural networks (CNNs), transformers) and tasks (language modeling, image classification, speech recognition) (Hestness et al., 2017; Kaplan et al., 2020; Rosenfeld et al., 2020; Alabdulmohsin et al., 2022; Sorscher et al., 2022; Cherti et al., 2023; Caballero et al., 2023). By fitting scaling exponents on midsize experiments, practitioners can forecast the dataset size or model size needed to reach a target accuracy without exhaustive searches (Hernandez et al., 2021; Hoffmann et al., 2022). For example, such methods guided the design of Chinchilla, which, with fewer parameters, outperformed much larger models like generative pre-trained transformer (GPT)-3 (Brown et al., 2020) by solving optimal model and data size using power-law relationship (Hoffmann et al., 2022). Scaling laws thus enable efficient allocation of resources when training state-of-the-art models.

Beyond their empirical utility, there is substantial theoretical interest in why scaling laws hold. A common theoretical formulation relates a model's generalization error to the training sample size $N$ via a power-law form $N^{-C/D}$. Here $D$ is a notion of data dimension and $C$ is a constant determined by properties of the target function that DNN models try to learn (Oono & Suzuki, 2019; Schmidt-Hieber, 2020). If data lie on a low-dimensional manifold, using its intrinsic dimension $H \ll D$ can yield faster rates than the ambient dimension (Nakada & Imaizumi, 2020; Liu et al., 2021; Dahal et al., 2022; Chen et al., 2022; Havrilla & Liao, 2024). These analyses – spanning fully-connected networks, CNNs, transformers – clarify how function class smoothness, model capacity, and data geometry together determine performance scaling.

Typically, the target function is assumed to belong to a smoothness class such as a Hölder (Oono & Suzuki, 2019; Schmidt-Hieber, 2020; Nakada & Imaizumi, 2020; Chen et al., 2022) or Besov (Liu et al., 2021) space with smoothness parameters and have an intrinsic dimension. The smoothness level

controls the constant $C$ (and sometimes exponent) in the generalization error bound $\widetilde{O}(N^{-C/D})$. In practice, one often estimates an effective scaling exponent empirically by training models on various dataset sizes and fitting a power-law curve. This curve guides extrapolation of dataset size and target performance on larger scales (Kaplan et al., 2020; Hoffmann et al., 2022).

However, smoothness alone may be too coarse to fully determine scaling behavior. In this paper, we investigate the role of local oscillatory complexity (LOC) on DNN scaling laws for data with fixed dimension. By LOC, we mean local variations in the target mapping that make it challenging for a network with spectral bias toward low frequencies to learn (Rahaman et al., 2019; Zhi-Qin et al., 2020). Parity functions (Daniely, 2017; Daniely & Malach, 2020; Kim & Suzuki, 2025) are an extreme example – highly oscillatory and notoriously hard for DNNs to learn – but our focus is on more gradated control of oscillations. Oscillatory behavior could arise, for example, in a system where multiple factors compensate each other while preserving some factors such as metabolic factors.

**Our contributions:**

- We introduce a metric based on the expected norm of its gradient to quantitatively measure a function's local oscillatory complexity. This metric captures the "up-and-down" variation of the target function on the data distribution.

- We propose a family of target functions with parity-like oscillations but continuous inputs. This surrogate retains the challenging high-frequency behavior of parity while being easier to learn and analyze in practice. We derive theoretical properties for this family: for any two functions with different frequencies, the standard generalization error bounds (which assume Hölder smoothness) differ only by constants regardless of frequencies. In log-log space, those error bounds would appear as vertically shifted curves.

- We conduct extensive experiments on learning these oscillatory target functions with feed-forward DNNs. We find that increasing oscillatory complexity can fundamentally alter scaling behavior. In particular, when the target is highly wiggly, the test loss scaling curve flattens out to an early saturation – adding more data yields little improvement, leading to no improvement in classification performance. Furthermore, comparing scaling curves across different oscillation levels, we observe that they are not mere shifts of one another – their shapes (exponents and curvature) differ. In fact, there appears to be a threshold of complexity beyond which the scaling exponent dramatically diminishes. We also compare these empirical findings to theoretical predictions and to power-law extrapolations. The classical theory would suggest scaling curves differ only by shifts; we find this holds only when oscillatory complexity differences are modest. When complexity varies greatly, a simple shift underestimates or overestimates performance.

In summary, data smoothness and dimension do not fully characterize for DNN scaling laws. Two target functions with the same Hölder exponent $\beta$ can exhibit very different scaling behavior if one is highly oscillatory. Our work opens the door to a sharper characterization of target functions – beyond traditional smoothness and dimension – that govern when and how DNN scaling laws break down.

The remainder of this paper is organized as follows. Sec. 2 provides background on generalization bounds, empirical scaling laws, and why parity functions are difficult for DNNs. Sec. 3 formalizes our problem statement. Sec. 4 introduces our oscillatory complexity measure, the parity-like target function $r(\mathbf{x})$, and potential scaling behavior for this function. Sec. 5 describes our experimental setup and results, demonstrating the effect of oscillatory complexity on scaling laws. We analyze the implications in Sec. 6, and conclude in Sec. 7.

## 2 BACKGROUND AND PRELIMINARIES

### 2.1 GENERALIZATION ERROR BOUNDS FOR DNNs

We begin by reviewing theoretical generalization error bounds for learning a target function with DNNs. Consider supervised learning with training data $\{\mathbf{x}_n, f(\mathbf{x}_n)\}_{n=1}^N$ where $\mathbf{x}_n \in \mathbb{R}^D$ are independent identically distributed (i.i.d.) samples and $f : \mathbb{R}^D \to \mathbb{R}$ is the target mapping. Let $\hat{f}^\star(\mathbf{x})$ be the empirical risk minimizer (ERM) to be obtained by solving the following optimization problem

during the DNN training

$$\hat{f}^{\star}(\mathbf{x}) = \arg\min_{\hat{f}(\mathbf{x})\in\mathcal{F}} \frac{1}{N} \sum_{n=1}^{N} \phi\left(\hat{f}(\mathbf{x}_n), f(\mathbf{x}_n)\right) \tag{1}$$

where $\mathcal{F}$ represents the DNN models with specific network architectures, and the $\phi(\cdot,\cdot)$ is the training loss function. The generalization error is the expected loss of $\hat{f}^{\star}(\mathbf{x})$ on the true target

$$\psi(f(\mathbf{x}), \hat{f}^{\star}(\mathbf{x})) = \mathbb{E}_{f(\mathbf{x})}\left[\sigma\left(f(\mathbf{x}), \hat{f}^{\star}(\mathbf{x})\right)\right] \tag{2}$$

where $\sigma(\cdot,\cdot)$ is a chosen loss (e.g. mean-squared error). As an example, for regression with squared loss and assuming $f(\mathbf{x})$ lies in a Hölder space with smoothness $\beta$, a representative bound from Schmidt-Hieber (2020) is

$$\psi(f(\mathbf{x}), \hat{f}^{\star}(\mathbf{x})) = \mathbb{E}_{f(\mathbf{x})}\left[\left(f(\mathbf{x}) - \hat{f}^{\star}(\mathbf{x})\right)^2\right] \leq C_1 N^{-\frac{2\beta}{2\beta+D}} L \log^2 N \tag{3}$$

for a constant $C_1$ when certain conditions on the DNN capacity are met. That is, when the depth $L$ of the DNNs in $\mathcal{F}$ is fixed, the error $\psi(f(\mathbf{x}), \hat{f}^{\star}(\mathbf{x}))$ is bounded by $\widetilde{O}(N^{-\frac{2\beta}{2\beta+D}})$, where $\widetilde{O}(\cdot)$ hides constants and logarithm factors.

If the data lies on an $H$-dimensional manifold (intrinsic dimension $H \ll D$), the rate improves. For instance, Nakada & Imaizumi (2020) give

$$\mathbb{E}_{f(\mathbf{x})}\left[\left(f(\mathbf{x}) - \hat{f}^{\star}(\mathbf{x})\right)^2\right] \leq C_2 N^{-\frac{2\beta}{2\beta+H}} (1 + \log N)^2 \tag{4}$$

The error bound $\psi(f(\mathbf{x}), \hat{f}^{\star}(\mathbf{x}))$ is on the order of $\widetilde{O}(N^{-\frac{2\beta}{2\beta+H}})$. Recently, Chen et al. (2022) derived a bound accounting for both ambient and intrinsic dimensions

$$\mathbb{E}_{f(\mathbf{x})}\left[\left(f(\mathbf{x}) - \hat{f}^{\star}(\mathbf{x})\right)^2\right] \leq C_3 \left(A^2 + \xi^2\right) \left(N^{-\frac{2\beta}{2\beta+H}} + \frac{D}{N}\right) \log^3 N \tag{5}$$

where $\|f(\mathbf{x})\|_{\infty} \leq A$, and $\xi$ is the variance proxy of noise in the data. Therefore, the error $\psi(f(\mathbf{x}), \hat{f}^{\star}(\mathbf{x}))$ is bounded by $\widetilde{O}\left(N^{-\frac{2\beta}{2\beta+H}} + \frac{D}{N}\right)$ taking $C_3\left(A^2 + \xi^2\right)$ as a constant.

Across these results, the Hölder smoothness parameter $\beta$, and the dimensions $(D, H)$ determine the exponent of $N$ in the bound. The Hölder radius $K$, which accounts for the LOC of the target, is absorbed into the constant factors (e.g. $C_1, C_2, C_3$) and does not affect the asymptotic rate $N^{-C/D}$.

## 2.2 MODELING SCALING LAWS WITH POWER-LAW FITS

In practice, researchers often empirically verify scaling laws by log-log plotting error versus resource and fitting a power-law. For a fixed DNN architecture, one commonly observes relationships of the form

$$l(N) := C_4 N^{-\alpha_1} \tag{6}$$
$$l(M) := C_5 M^{-\alpha_2} \tag{7}$$
$$l(U) := C_6 U^{-\alpha_3} \tag{8}$$

for test loss $l$ as a function of training set size $N$, model parameters $M$, or compute $U$. Here $C_4, C_5, C_6$ are prefactors and $\alpha_1, \alpha_2, \alpha_3$ are the fitted exponents. Kaplan et al. (2020) popularized this approach for language models, demonstrating tight power-law fits over broad scales. Given a few experimental points, one can extrapolate performance to larger $N, M, U$ without exhaustive sweeps.

Hoffmann et al. (2022) extended this to two-variable scaling and introduced irreducible loss $B$ to handle plateaus. For example, they fit a form

$$l(N, M) := B + C_8 N^{-\alpha_4} + C_9 M^{-\alpha_5} \tag{9}$$

where $B$ is the asymptotic loss floor, $C_8$ and $C_9$ are multiplicative constants, and $\alpha_4, \alpha_5$ capture data and model scaling respectively. Given a fixed compute budget $U$, such fits inform the optimal split

between data and model size for training. Overall, the key content of these empirical models lies in the fitted exponents and prefactors. Different papers may use slightly different fitting functions or include extra terms, but they all boil down to estimating these parameters.

When focusing on test loss versus data size $N$ (with model fixed), the various forms above can be simplified. One can write

$$l(N) := B + C_8 N^{-\alpha_4} \tag{10}$$

as a general functional form. Fitting such a model can be done by solving a regression problem in log-space using robust fitting like Huber loss $\zeta_\theta(\cdot)$ (Huber, 1964; Hoffmann et al., 2022)

$$\min_{c_8, \alpha_4, b} \frac{1}{W} \sum_{w=1}^{W} \zeta_\theta \left( \log_a l(N_w) - \log_a \left( a^{c_8 - \alpha_4 \log_a N_w} + a^b \right) \right) \tag{11}$$

where the DNN model is trained with varying data size $\{N_w\}_{w=1}^{W}$, and $C_8 = a^{c_8}, B = a^b$. The (11) can be solved for a local minima by choosing the best fit from multiple initializations of parameters using Broyden–Fletcher–Goldfarb–Shanno (BFGS) algorithm (Nocedal, 1980).

**Scaling laws vs. theory:** There are emerging theoretical links between empirical scaling exponents and structural properties of models and data. For instance, in kernel or infinite-width network analyses, the decay of kernel eigenvalues can determine the effective data-scaling exponent. Specifically, if the eigenvalues $\lambda_k$ follow a power-law tail $\lambda_k \propto k^{-(1+\alpha_6)}$, then in the resolution-limited regime the test loss scales as $N^{-\alpha_6}$ up to logarithmic factors. In this setting, the empirical exponent $\alpha_1$ observed in practice can be identified with the theoretical spectral exponent $\alpha_6$ (Bahri et al., 2024). This connection shows that classical smoothness and dimension parameters, which shape eigenvalue decay, may reappear in practice as fitted scaling exponents. Consequently, scaling experiments allow us to infer aspects of the target function's effective smoothness and dimensionality through the fitted exponent, even though $\beta$ and $D$ (or $H$) are not directly observable.

In our work, we will compare what theory *would predict* for varying oscillation (namely, only a shift in the curve, not a change in $\alpha_4$) to what we actually *observe* empirically (which will show changes in the fitted $\alpha_4$ for different oscillation levels).

## 2.3 WHY PARITY FUNCTIONS ARE HARD FOR DNNs

A canonical example of a highly oscillatory target is the parity function. Let $g : \{0,1\}^D \to \{-1,1\}$ be a parity function defined by

$$g(\mathbf{x}) := (-1)^{\mathbf{x}^\top \mathbf{v}}, \tag{12}$$

where $\mathbf{v} \in \{0,1\}^D$. $g(\mathbf{x})$ outputs 1 or $-1$ depending on whether the sum of the selected input bits (those where $v_d = 1$) is even or odd. Parity is extremely wiggly: flipping any one input bit in the support $\mathcal{S} := \{d \in \{1, \ldots, D\} | v_d \neq 0\}$ flips the output. Thus, parity's Fourier spectrum is concentrated on high-frequency components. Standard neural networks have a known spectral bias: they tend to learn low-frequency (smooth) functions first and struggle with high-frequency functions (Rahaman et al., 2019; Zhi-Qin et al., 2020). This makes parity notoriously difficult for gradient-based training. Indeed, Shalev-Shwartz et al. (2017) showed that for learning tasks drawn from a large family of orthogonal functions, which is the case for parity functions with different $\mathbf{v}$, gradient descent provides little target-specific signal – gradients are almost independent of which particular function in the family one is trying to learn.

Additionally, while specialized methods can solve parity under certain conditions (e.g. input data distribution (Dahal et al., 2022), transformers and reasoning techniques (Han & Ghoshdastidar, 2025; Kim & Suzuki, 2025) or certain DNN initializations (Abbe et al., 2025)), these are more like exceptions than the rule. In general, parity represents an extreme case of oscillation that breaks many of the assumptions under which DNNs perform well.

Because parity is so challenging and oscillatory, it serves as a useful testbed for our question: does a function's local oscillatory complexity affect not only whether it can be learned, but how the learning scales with data? In other words, even if we eventually can learn a wiggly function given enough data and a large network, do we observe a different scaling law compared to a smoother function? This is precisely what we explore, using a parity-like function $r(\mathbf{x})$ defined on $\mathbb{R}^D$ as described in Sec. 4.2.

## 3 PROBLEM STATEMENT

We study how the scaling behavior of DNNs is affected by the LOC of the target function. Our focus is on parity-like target functions that exhibit strong local oscillations. Concretely, we pose three key questions:

- **(Q1)** Does a highly oscillatory target function materially affect the empirical scaling behavior of DNNs (test loss vs. training data size)?
- **(Q2)** If yes, how does this effect manifest? Is it simply a vertical shift of the scaling curve (i.e. a higher error floor), a change in the power-law exponent, the appearance of distinct scaling regimes or saturation effects, or some combination of these?
- **(Q3)** Do existing theoretical generalization bounds and practical power-law extrapolation methods accurately predict the scaling behavior when the target is very oscillatory, or do they break down?

## 4 METHODOLOGY

### 4.1 QUANTIFYING LOCAL OSCILLATORY COMPLEXITY

We define a metric to quantify how *locally oscillatory* a function is, which intuitively measures how much the function output wiggles up and down in local regions of input space. A natural choice is the expected magnitude of the function's gradient, since large gradients indicate rapid changes. Formally, for a target differentiable function $f(\mathbf{x})$ and an input distribution $p(x)$, we define

$$\eta(f(\mathbf{x})) := \mathbb{E}_{p(\mathbf{x})}[\|\nabla f(\mathbf{x})\|_2] \tag{13}$$

This captures the average local variation of $f$ under the data distribution. In practice, we focus on the local oscillation on the test distribution, since generalization performance is directly impacted by how the function behaves on test data. Thus, we estimate $\eta(f)$ empirically using held-out test samples

$$\hat{\eta}(f(\mathbf{x})) := \frac{1}{I} \sum_{i=1}^{I} \|\nabla f(\mathbf{x}_i)\|_2 \tag{14}$$

for test points $\{\mathbf{x}_i\}_{i=1}^{I}$. In our experiments, $\hat{\eta}(f)$ provides a single scalar complexity measure for each target function instance.

This metric effectively averages the local slope of the function. A function with many rapid oscillations (high-frequency content) will have a larger gradient norm on average. Note that $\eta(f)$ is related to the Lipschitz constant of $f$, but instead of a worst-case maximum, it's an average magnitude under the data distribution. It is also related to the Sobolev or Besov smoothness norms, but we treat it as a more direct empirical measure of LOC.

### 4.2 PARITY-LIKE OSCILLATORY TARGET FUNCTION

To systematically study the effect of oscillations, we consider a target function family where oscillatory complexity is *controllable*. The strict parity function $g(\mathbf{x})$ is too extreme. Instead, we use a continuous surrogate that retains parity's oscillatory flavor. Specifically, we define

$$r(\mathbf{x}) := \cos(\omega\|\mathbf{x}\|_1) \tag{15}$$

where $\omega > 0$ is a frequency parameter.

This choice has several advantages. First, when $\mathbf{x}$ has binary components and $\omega$ is an odd multiple of $\pi$, $r(\mathbf{x})$ reduces to the parity function by Euler's formula. Thus, parity is a special case of $r(\mathbf{x})$. Second, by varying $\omega$, we can smoothly control the oscillation frequency: larger $\omega$ means $r(\mathbf{x})$ oscillates more rapidly as a function of $\mathbf{x}$. Third, unlike strict parity, $r(\mathbf{x})$ is differentiable (subgradient at $\mathbf{0}$) and its frequency components are not orthogonal (which actually helps gradient-based learning). In fact, $\|\nabla r(\mathbf{x})\|_2 = \omega D^{\frac{1}{2}}|\sin(\omega\|\mathbf{x}\|_1)|$, which shows how $\omega$ influences local variation directly. Overall, $r(\mathbf{x})$ behaves like a parity function in terms of LOC, but is easier for standard networks to learn (especially at moderate $\omega$). We use $r(\mathbf{x})$ as our target function in all experiments, treating $\omega$ (or

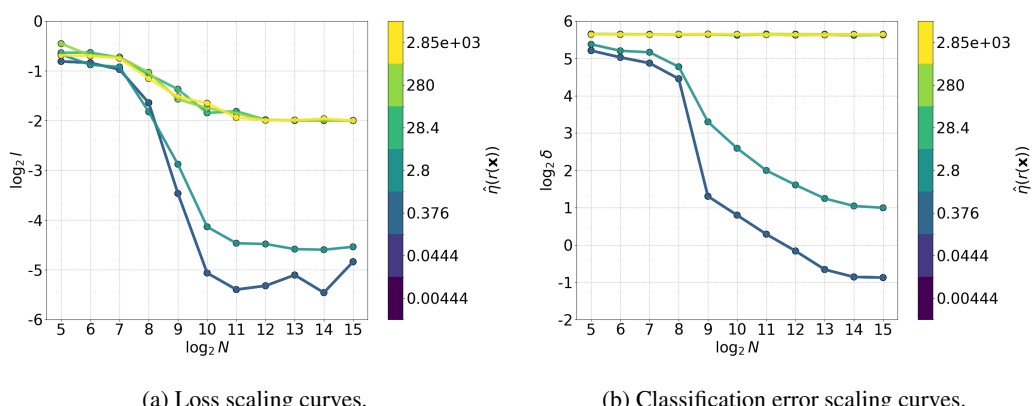

(a) Loss scaling curves.                    (b) Classification error scaling curves.

Figure 1: Scaling curves.

equivalently $\hat{\eta}(r)$) as the knob for oscillatory complexity. The Hölder class of $r(\mathbf{x})$ does not change with $\omega$ (proof in the Appendix A.1) but the Hölder radius does.

In one dimension,

$$r'(x) = -\omega \, \sin(\omega|x|) \, \mathrm{sign}(x) \quad (x \neq 0), \qquad |r'(x)| \leq \omega, \tag{16}$$

and $r$ is Lipschitz with constant $\omega$. Thus $r \in \mathcal{C}^{0,1}$ with Hölder radius $K(r) \asymp \omega$, but $r \notin \mathcal{C}^\beta$ for any $\beta > 1$ due to the cusp at $x = 0$. In $D$ dimensions with $r(\mathbf{x}) = \cos(\omega\|\mathbf{x}\|_1)$, for $\mathbf{x} \neq \mathbf{0}$,

$$\nabla r(\mathbf{x}) = -\omega \, \sin(\omega\|\mathbf{x}\|_1) \, \mathrm{sign}(\mathbf{x}), \qquad \|\nabla r(\mathbf{x})\|_2 \leq \omega\sqrt{D}, \tag{17}$$

so the same conclusion holds: $r \in \mathcal{C}^{0,1}$ with $K \asymp \omega$, but not in $\mathcal{C}^{\beta>1}$. Consequently, changing $\omega$ leaves the theoretical $N$-exponent for $\beta = 1$ intact, while increasing the approximation budget through the Hölder radius $K \sim \omega$.

# 5 EXPERIMENTS

## 5.1 SETUP

**Data generation:** We construct a binary classification task based on the target function $r(\mathbf{x})$. For each trial, we fix an input dimension $D$ and frequency $\omega$. We sample input vectors $\mathbf{x} \sim \mathcal{N}(\mathbf{0}, \mathbf{I})$. We then compute $y = r(\mathbf{x})$ and binarize the output by thresholding at the median value, yielding labels in $\{0, 1\}$ (approximately balanced classes). This procedure ensures that the classification boundary is implicitly defined by the oscillatory function $r(\mathbf{x})$.

We vary the frequency $\omega$ over a range spanning low to high oscillation. Specifically, $\omega$ is swept through $10^{-3}\pi, 10^{-2}\pi, 10^{-1}\pi, 10^0\pi, 10^1\pi, 10^2\pi, 10^3\pi$. This produces target functions $r(\mathbf{x})$ with a wide range of $\hat{\eta}(r(\mathbf{x}))$ values, from very smooth to highly wiggly. We also consider input dimensions $D \in \{1, 2, 4\}$ to see the effect of data dimensionality.

For each $(D, \omega)$ condition, we generate a dataset and then subsample training sets of various sizes $N$ to trace a scaling curve. We use $N \in \{2^5, 2^6, 2^7, ..., 2^{15}\}$. We hold out a validation set and a test set, each of size $10,000$ examples, for early stopping and final evaluation.

**Model and training:** We use a fixed multi-layer perceptron (MLP) architecture for all experiments. Since our focus is on data scaling, we keep the model fixed. We train with mean-squared-error (MSE) loss to predict labels under various $(D, \omega)$ conditions. Details of DNN architecture and training are in the Appendix A.2.

## 5.2 RESULTS

**(Q1): Does LOC affect scaling? – Yes.**

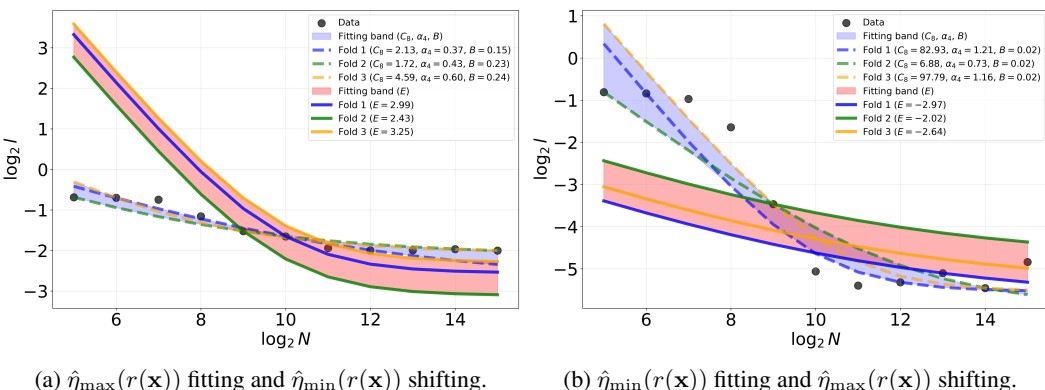

(a) $\hat{\eta}_{\max}(r(\mathbf{x}))$ fitting and $\hat{\eta}_{\min}(r(\mathbf{x}))$ shifting.  (b) $\hat{\eta}_{\min}(r(\mathbf{x}))$ fitting and $\hat{\eta}_{\max}(r(\mathbf{x}))$ shifting.

Figure 2: Fitted curves.

Fig. 1(a) shows test loss vs. training size for varying local oscillatory complexity with dimension $D = 2$ (figures for other dimensions are attached in the Appendix A.3). As $\hat{\eta}(r(\mathbf{x}))$ increases (higher $\omega$), the scaling curves shift upward and compress. In particular, high-complexity targets have significantly higher error floors and smaller ranges of improvement. For the most wiggly functions, the test loss drops by less than two orders of magnitude across $N$, indicating an early onset of the irreducible error (saturation). In contrast, low-complexity targets continue to improve steadily with more data. In other words, when the target function is very oscillatory, adding more data yields diminishing returns – the model struggles to convert additional samples into significantly lower error. This directly answers **(Q1):** higher LOC does materially hurt and limit scaling performance.

**Remark.** While the test loss continues to decrease with larger $N$ under high-complexity targets, it quickly saturates, and the classification accuracy $\delta$ in Fig. 1(b) shows virtually no improvement. This observation challenges a common assumption in the community: that simply increasing the dataset size invariably leads to better DNN performance. Our results indicate that at a certain turn-off frequency, scaling fails to occur. This highlights an important implication for future practice: beyond model and data size, intrinsic data properties must also be accounted when forecasting scaling behavior or planning large-scale training.

**Q2: How does the effect manifest? – It changes both exponent and prefactor of the scaling law, not just a simple shift.**

In Fig. 1(a), we observe that the high-frequency (high $\hat{\eta}(r(\mathbf{x}))$) curves not only lie above the low-frequency ones, but also tend to flatten in slope. For example, the curve for $\omega = 10^3 \pi$ (very high complexity) almost plateaus, whereas the curve for $\omega = 10^{-3}\pi$ has a steep slope through the range shown. A similar pattern holds for other dimensions (See figures in the Appendix A.3): low $\omega$ curves drop rapidly (steep slope) and maintain power-law behavior longer, while high $\omega$ curves level off sooner.

Fig. 1(a) also shows that the scaling curves are not simple shifts of one another. Only when comparing among the most extreme frequencies (e.g., within a cluster of all very high $\omega$ curves) do differences between curves become nearly zero. But across moderate vs. high frequencies, the differences are pronounced.

**Q3: Do theory and standard extrapolation hold? – Only partially.**

Theoretical bounds based on Hölder class would suggest the curves in Fig. 1(a) (also for other dimensions) differ by a constant factor. This does not hold across the full range of $\omega$, as we have discussed in the **(Q2)**. The practical power-law extrapolation also fails when complexity changes substantially. We attempted to use a power-law fit on a baseline (either lowest-$\hat{\eta}_{\min}(r(\mathbf{x}))$ or highest-$\hat{\eta}_{\max}(r(\mathbf{x}))$) scaling curve by estimating $C_8, \alpha_4, B$ in (11), then shift it with an additive constant $E$ to predict the other scaling curves (either $\hat{\eta}_{\max}(r(\mathbf{x}))$ or $\hat{\eta}_{\min}(r(\mathbf{x}))$). This step mimics the theoretical assumption of proportional error plus a shift in log-space. All estimations are performed with 3-fold cross-validation. Details of curve fitting are attached in the Appendix A.4.

Table 1: Power-law fitting and baseline shifting performance.

| $\omega$ | $\hat{\eta}(r(\mathbf{x}))$ | Power law | | | $\hat{\eta}_{\max}(r(\mathbf{x}))$ shift | | | $\hat{\eta}_{\min}(r(\mathbf{x}))$ shift | | |
| --- | --- | --- | --- | --- | --- | --- | --- | --- | --- | --- |
| | | MAE | RMSE | HL | MAE | RMSE | HL | MAE | RMSE | HL |
| $10^{-3}\pi$ | 0.0044 | 0.72 | 0.80 | 0.0025 | 1.4 | 1.6 | 0.0052 | —[1] | — | — |
| $10^{-2}\pi$ | 0.044 | 0.72 | 0.80 | 0.0025 | 1.4 | 1.6 | 0.0052 | 0.68 | 0.76 | 0.0024 |
| $10^{-1}\pi$ | 0.38 | 0.72 | 0.80 | 0.0025 | 1.4 | 1.6 | 0.0052 | 0.68 | 0.76 | 0.0024 |
| $10^{0}\pi$ | 2.8 | 0.57 | 0.72 | 0.0020 | 1.1 | 1.3 | 0.0041 | 0.56 | 0.69 | 0.0019 |
| $10^{1}\pi$ | 28 | 0.21 | 0.23 | 0.00075 | 0.17 | 0.19 | 0.00060 | 1.3 | 1.6 | 0.0047 |
| $10^{2}\pi$ | 280 | 0.15 | 0.17 | 0.00055 | 0.17 | 0.18 | 0.00064 | 1.4 | 1.6 | 0.0048 |
| $10^{3}\pi$ | 2800 | 0.19 | 0.22 | 0.00071 | — | — | — | 1.3 | 1.7 | 0.0047 |

Fig. 2 illustrates the curves by power-law fitting and baseline shifting for $D = 2$ (Other figures are shown in Appendix A.5). The shifted curves (red band) systematically misestimate the true behavior: shifting from a $\hat{\eta}_{\min}(r(\mathbf{x}))$ baseline to a $\hat{\eta}_{\max}(r(\mathbf{x}))$ target leads to severe underestimation at small $N$ and overestimation at large $N$, whereas shifting a $\hat{\eta}_{\max}(r(\mathbf{x}))$ baseline to a $\hat{\eta}_{\min}(r(\mathbf{x}))$ target does the opposite. This is because the target curve has a different curvature that a mere shift can't capture. We quantified the fitting error (mean absolute error (MAE), root mean square error (RMSE), Huber loss (HL)) for both direct power-law fits and shifted fits in Tables 1 (Other tables are shown in Appendix A.6). Notably, when $\hat{\eta}(r(\mathbf{x}))$ differs substantially from the baseline $\hat{\eta}_{\max}(r(\mathbf{x}))$ or $\hat{\eta}_{\min}(r(\mathbf{x}))$, the direct power-law fit on each curve yields lower error than trying to use a shift from baseline fits. Indeed, the error of the shifted approximation jumps dramatically when oscillatory complexities differ beyond a certain point (e.g. comparing $\hat{\eta}(r(\mathbf{x})) \approx 2.8$ vs. $\hat{\eta}(r(\mathbf{x})) \approx 28$).

These results are anticipated in Schmidt-Hieber (2020) and related works on ReLU approximation of Hölder functions (Yarotsky, 2017; 2018), which show that if $f \in \mathcal{C}^{0,1}([0,1]^t, K)$ (the Lipschitz/Hölder class with $\beta = 1$ and radius $K$), then for ReLU networks of sparsity $s$ and logarithmic depth, one has the uniform approximation bound

$$\inf_{g \in \mathcal{F}(L,p,s,F)} \|f - g\|_\infty \lesssim K \, s^{-1/t}.$$

Thus the exponent in $s$ depends only on $\beta$ and $t$, but the Hölder radius $K$ multiplies the error. For our surrogate $r(\mathbf{x}) = \cos(\omega\|\mathbf{x}\|_1)$ we have $\beta = 1$ and $K \asymp \omega$ (global Lipschitz). Consequently, increasing frequency $\omega$ leaves the statistical $N$-rate exponent unchanged but linearly inflates the approximation budget: to achieve tolerance $\varepsilon$ one requires $s \gtrsim (\omega/\varepsilon)^t$. If $s$ is held fixed, larger $K$ lowers the crossover point $N_\star$ where approximation error dominates, explaining the earlier saturation of high-LOC curves observed in our experiments.

Oscillatory complexity affects all parameters of the scaling law – the exponent, the prefactor, and even the additive residual. Existing theoretical bounds, which absorb complexity into constants and imply scaling curves should just shift, are valid only when oscillation levels are similar. Empirically, when complexity exceeds a threshold, those bounds become loose and a new power-law regime emerges. Meanwhile, practical power-law fitting inherently accounts for whatever oscillation is present in the data (since it fits each case separately), but it offers no guarantee for generalization if the complexity differs between training and deployment data. In fact, if a model were trained on a low-oscillation dataset and tested on a high-oscillation scenario, our results indicate its performance could be much worse than predicted by simply extrapolating the training scaling law. This underscores the importance of considering local oscillatory complexity as a factor in generalization.

## 6    DISCUSSION

**Scaling law behavior depends on target function complexity.** Our findings highlight that the commonly observed power-law scaling of DNN performance can break down when the target function is highly oscillatory. In classical smoothness terms, all our target functions $r(\mathbf{x})$ are equally smooth ($\beta = 1$), yet their scaling behavior differs markedly. This indicates that oscillatory complexity is a distinct aspect of function complexity not captured by $\beta$. High oscillation introduces an effective

---

[1]No shifting between identical local oscillatory complexities.

data inefficiency: beyond a certain sample size, the model cannot improve much because it struggles with the fine-grained patterns in the target. This manifests as an elevated "irreducible" error floor and a smaller effective exponent.

**Scaling regime changes with high oscillatory targets.** In our experiments, highly oscillatory targets often produced regime changes in the scaling curves. Instead of following a long power-law decline, the curves flatten almost immediately. This indicates that when oscillations dominate, scaling effectively fails—additional data yields little or no benefit, especially in the classification performance. Such behavior underscores the need to consider data properties, not just size, when predicting the scalability of deep models.

**Theoretical implications:** Our experiments confirm that data dimensionality and Hölder class continuity alone do not fully determine theoretical bounds. When the LOC varies substantially, it appears to influence the scaling laws in a nontrivial way. Specifically, the empirical scaling exponent $\alpha_4$ (Eq. (10)) depends on $\hat{\eta}(r(\mathbf{x}))$, even when $\beta$ remains fixed across different values of $\omega$. Indeed, our DNNs yielded distinct $\alpha_4$ for different $\omega$. This arises because the Hölder radius $K$ is governed by the LOC: when LOC is large, approximation error dominates the scaling behavior. These findings suggest that LOC is a critical determinant of scaling laws, shaping not only the specific exponents but also whether meaningful scaling persists or ultimately breaks down.

**Practical considerations:** For practitioners, an important takeaway is that extrapolating a scaling law beyond the regime of oscillatory complexity it was fit on can lead to error. If one's training data is much less complex (smoother) than the possible test scenarios, a model might appear to scale well (following a steep power law) on the training distribution, yet perform worse on a more complex test distribution than predicted by simply adding more data of the same smoothness. In other words, data quality in terms of target complexity matters: simply collecting more samples may not close the gap if the new scenario is more complex. One might need additional model capacity or fundamentally different approaches (e.g. architectures or features that capture the oscillatory structure) to improve in that case. Conversely, if test data is smoother than training data, the model might be over-prepared (having learned to fit unnecessary wiggles).

## 7 CONCLUSION

We presented a systematic study of how target function LOC impacts the scaling laws of deep neural networks. High oscillatory complexity can severely limit the benefits of additional training data, causing early saturation of test performance. Our results answered three key questions: **(Q1)** LOC materially affects scaling behavior, often leading to higher error floors. **(Q2)** This effect is not merely a vertical shift in the log-log scaling plot, but a change in the apparent empirical exponent and scaling regimes. **(Q3)** Classical generalization bounds, which absorb the Hölder radius $K$ into constants, do not contradict these findings: in theory the asymptotic rate is unchanged, but larger $K$ inflates approximation constants and lowers the crossover point where approximation error dominates. Empirically, this shift in regimes manifests itself as a change in the fitted exponent, even though the underlying theoretical exponent remains fixed. Oscillatory complexity does not alter the asymptotic exponent itself, however, it critically determines when and how scaling laws break down in practice.

These findings emphasize that smoothness class alone is insufficient to characterize scaling behavior; the oscillatory nature of the target plays a crucial role in how a DNN's error scales with data. For theory, this suggests that new parameters—related to higher-order derivatives or frequency spectra—should be incorporated into generalization bounds. In practice, researchers should be cautious when extrapolating scaling laws beyond regimes of similar complexity and should consider methods that explicitly address highly oscillatory targets.

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

# A APPENDIX

## A.1 PROOFS

Given the target mapping $r(\mathbf{x})$ in (15), we show that its Hölder class is unaffected by $\omega$

**Proof:** *We first show that the smoothness parameter $\beta$ is independent of frequency parameter $\omega$.*
*$\forall \mathbf{x}, \mathbf{y} \in \mathbb{R} \setminus \{\mathbf{0}\}$,*

$$|r(\mathbf{x}) - r(\mathbf{y})| = |\cos(\omega \|\mathbf{x}\|_1) - \cos(\omega \|\mathbf{y}\|_1)| \le |\omega| \, |\|\mathbf{x}\|_1 - \|\mathbf{y}\|_1| \le |\omega| \|\mathbf{x} - \mathbf{y}\|_1$$
$$\le |\omega| D \, \|\mathbf{x} - \mathbf{y}\|_\infty \qquad (18)$$

*From (18) we obtain that $r(\mathbf{x})$ is Lipschitz with constant $|\omega| D$. Also $\|r(\mathbf{x})\|_\infty \le 1$, so we may take a Hölder radius $K \ge max(1, |\omega| D)$ and conclude $r(\mathbf{x}) \in \mathcal{C}^{0,1}$ (i.e. $\beta = 1$).*

*Therefore, the bounds in (3)– (5) keep the same $N$-exponent but differ by the multiplicative constants that depend on $\omega$ (through the Hölder radius $K$).*

## A.2 DNN ARCHITECTURE AND TRAINING

The DNN is a multilayer perceptron with hidden-layer sizes $\{1024, 1024, 1024, 1024, 1024, 1024, 1024, 1024, 1024, 512, 256, 128, 64, 32, 16, 8, 4, 2\}$. Each hidden layer is followed by batch normalization and a ReLU activation, except the final layer. This architecture is chosen to be sufficiently expressive yet otherwise generic; it was not tailored to the parity task. The specific widths are not critical, as our study focuses on scaling behavior with training data size rather than model design.

Training uses the Adam optimizer with a learning rate of $10^{-4}$ and a mini-batch size of 128. Early stopping based on validation accuracy is applied. For each condition, we train with five random initializations and report the average test performance.

## A.3 SCALING CURVES

Fig. 3 and 4 present the scaling curves of test loss and classification error as the training data size varies across all dimensions.

## A.4 SCALING CURVE FITTING

We fit power-law models to the scaling curves in Fig. 1(a) by solving (11). The parameters $b$, $c_8$, and $\alpha_4$ were initialized as $\log_2(l(N_{\max}))$, $\log_2(l(N_{\min}) - l(N_{\max}))$, and 0.5, respectively, following the same scale as in Hestness et al. (2017); Bahri et al. (2024); Rosenfeld et al. (2020); Sharma & Kaplan (2022), with additional Gaussian noise added. We then performed 3-fold cross-validation on the full set of 11 $(\log_2 N, \log_2 l)$ pairs for each curve. For each fold, 100 random initializations of (11) were executed with a fixed $\theta = 10^{-3}$, and the fit achieving the lowest Huber loss on the training split was selected.

To further compare empirical fits with theoretical generalization bounds, we carried out an additional fitting procedure. To test if theoretical error bounds differ by shifts in logarithmic scale under varying $\omega$, we introduced an additional shift parameter $E$. Specifically, for each dimension $D$, we used the fitted parameters $c_8$, $\alpha_4$, and $b$ obtained from scaling curves with $\hat{\eta}_{\max}(r(\mathbf{x}))$ and $\hat{\eta}_{\min}(r(\mathbf{x}))$ across 3 folds, yielding 6 baseline parameter sets. Given each baseline, we solved the following optimization problem to estimate only the shift $E$:

$$\min_E \frac{1}{W} \sum_{w=1}^{W} \zeta_\theta \left( \log_a l(N_w) - \log_a \left( a^{c_8 - \alpha_4 \log_a N_w} + a^b \right) - E \right) \qquad (19)$$

This procedure was again applied with 3-fold cross-validation for each scaling curve under different $\hat{\eta}(r(\mathbf{x}))$, with $E$ initialized at 0.

## A.5 FITTED CURVES

Fig. 5 and 6 present the fitted curves obtained from power-law fitting and baseline shifting across different dimensions. In these figures, the baseline parameters $c_8$, $\alpha_4$, and $b$ are always taken from

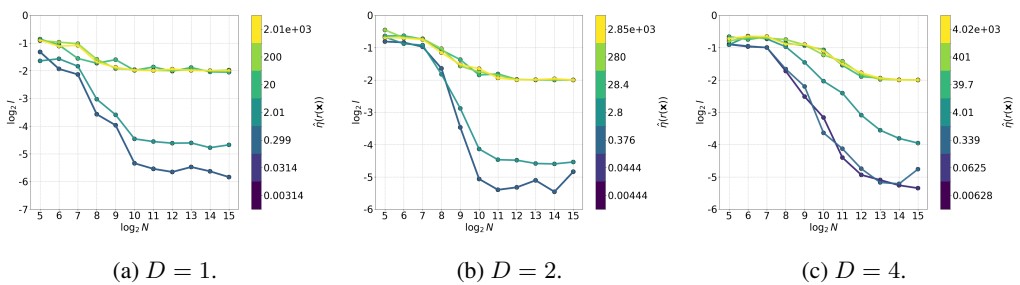

Figure 3: Test loss scaling curves.

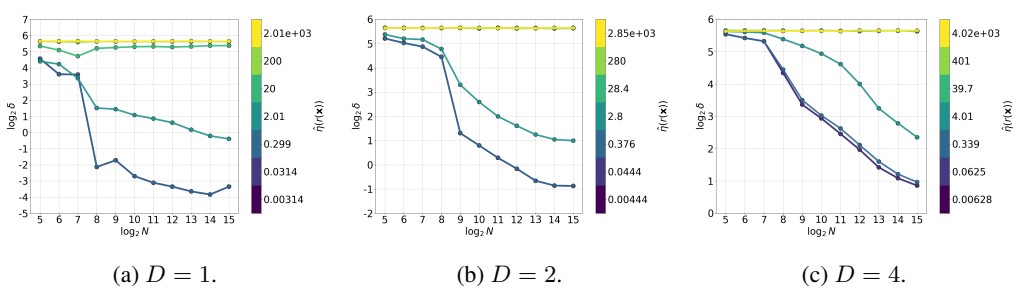

Figure 4: Classification error scaling curves.

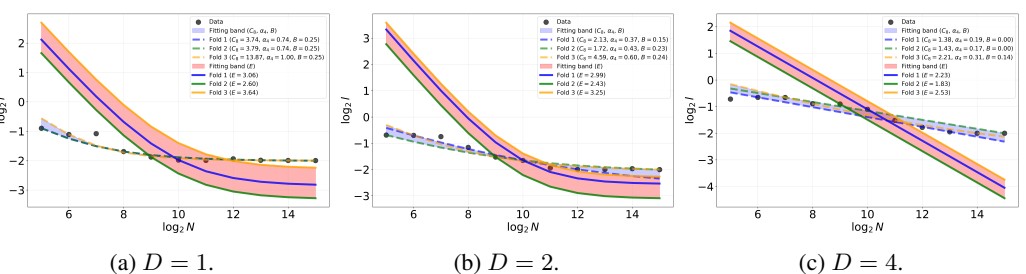

Figure 5: Fitted curves for $\hat{\eta}_{\max}(r(\mathbf{x}))$ fitting and $\hat{\eta}_{\min}(r(\mathbf{x}))$ shifting.

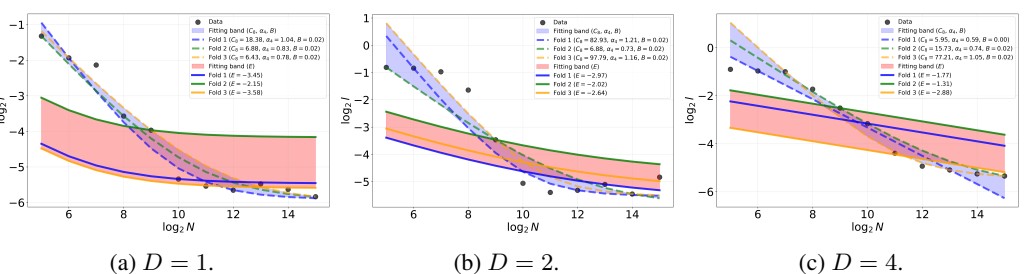

Figure 6: Fitted curves for $\hat{\eta}_{\min}(r(\mathbf{x}))$ fitting and $\hat{\eta}_{\max}(r(\mathbf{x}))$ shifting.

Table 2: Power-law fitting and baseline shifting performance ($D = 1$).

| $\omega$ | $\hat{\eta}(r(\mathbf{x}))$ | Power law | | | $\hat{\eta}_{\max}(r(\mathbf{x}))$ shift | | | $\hat{\eta}_{\min}(r(\mathbf{x}))$ shift | | |
|---|---|---|---|---|---|---|---|---|---|---|
| | | MAE | RMSE | HL | MAE | RMSE | HL | MAE | RMSE | HL |
| $10^{-3}\pi$ | 0.0031 | 0.34 | 0.41 | 0.0012 | 1.4 | 1.7 | 0.0052 | — | — | — |
| $10^{-2}\pi$ | 0.031 | 0.34 | 0.41 | 0.0012 | 1.4 | 1.7 | 0.0052 | 0.29 | 0.34 | 0.0011 |
| $10^{-1}\pi$ | 0.30 | 0.34 | 0.41 | 0.0012 | 1.4 | 1.7 | 0.0052 | 0.29 | 0.34 | 0.0011 |
| $10^{0}\pi$ | 2.0 | 0.30 | 0.40 | 0.0011 | 0.97 | 1.2 | 0.0035 | 0.47 | 0.57 | 0.0016 |
| $10^{1}\pi$ | 20 | 0.12 | 0.15 | 0.00043 | 0.096 | 0.11 | 0.00036 | 1.3 | 1.5 | 0.0045 |
| $10^{2}\pi$ | 200 | 0.19 | 0.25 | 0.00068 | 0.11 | 0.16 | 0.00041 | 1.3 | 1.5 | 0.0046 |
| $10^{3}\pi$ | 2000 | 0.11 | 0.16 | 0.00038 | — | — | — | 1.3 | 1.5 | 0.0047 |

Table 3: Power-law fitting and baseline shifting performance ($D = 2$).

| $\omega$ | $\hat{\eta}(r(\mathbf{x}))$ | Power law | | | $\hat{\eta}_{\max}(r(\mathbf{x}))$ shift | | | $\hat{\eta}_{\min}(r(\mathbf{x}))$ shift | | |
|---|---|---|---|---|---|---|---|---|---|---|
| | | MAE | RMSE | HL | MAE | RMSE | HL | MAE | RMSE | HL |
| $10^{-3}\pi$ | 0.0044 | 0.72 | 0.80 | 0.0025 | 1.4 | 1.6 | 0.0052 | — | — | — |
| $10^{-2}\pi$ | 0.044 | 0.72 | 0.80 | 0.0025 | 1.4 | 1.6 | 0.0052 | 0.68 | 0.76 | 0.0024 |
| $10^{-1}\pi$ | 0.38 | 0.72 | 0.80 | 0.0025 | 1.4 | 1.6 | 0.0052 | 0.68 | 0.76 | 0.0024 |
| $10^{0}\pi$ | 2.8 | 0.57 | 0.72 | 0.0020 | 1.1 | 1.3 | 0.0041 | 0.56 | 0.69 | 0.0019 |
| $10^{1}\pi$ | 28 | 0.21 | 0.23 | 0.00075 | 0.17 | 0.19 | 0.00060 | 1.3 | 1.6 | 0.0047 |
| $10^{2}\pi$ | 280 | 0.15 | 0.17 | 0.00055 | 0.17 | 0.18 | 0.00064 | 1.4 | 1.6 | 0.0048 |
| $10^{3}\pi$ | 2800 | 0.19 | 0.22 | 0.00071 | — | — | — | 1.3 | 1.7 | 0.0047 |

Table 4: Power-law fitting and baseline shifting performance ($D = 4$).

| $\omega$ | $\hat{\eta}(r(\mathbf{x}))$ | Power law | | | $\hat{\eta}_{\max}(r(\mathbf{x}))$ shift | | | $\hat{\eta}_{\min}(r(\mathbf{x}))$ shift | | |
|---|---|---|---|---|---|---|---|---|---|---|
| | | MAE | RMSE | HL | MAE | RMSE | HL | MAE | RMSE | HL |
| $10^{-3}\pi$ | 0.0063 | 0.66 | 0.78 | 0.0024 | 1.5 | 1.6 | 0.0055 | — | — | — |
| $10^{-2}\pi$ | 0.062 | 0.66 | 0.78 | 0.0024 | 1.5 | 1.6 | 0.0055 | 0.52 | 0.56 | 0.0019 |
| $10^{-1}\pi$ | 0.34 | 0.76 | 0.90 | 0.0028 | 1.4 | 1.5 | 0.0050 | 0.68 | 0.72 | 0.0026 |
| $10^{0}\pi$ | 4.0 | 0.37 | 0.45 | 0.0013 | 0.72 | 0.78 | 0.0027 | 0.57 | 0.77 | 0.0020 |
| $10^{1}\pi$ | 40 | 0.27 | 0.30 | 0.0010 | 0.21 | 0.23 | 0.00079 | 1.3 | 1.4 | 0.0046 |
| $10^{2}\pi$ | 400 | 0.22 | 0.26 | 0.00083 | 0.21 | 0.24 | 0.00076 | 1.3 | 1.5 | 0.0048 |
| $10^{3}\pi$ | 4000 | 0.24 | 0.28 | 0.00087 | — | — | — | 1.3 | 1.5 | 0.0047 |

the first fold of the cross-validation during power-law fitting. The additional shift parameter $E$ is then estimated using 3-fold cross-validation. Note that shifting is performed only between $\hat{\eta}_{\max}(r(\mathbf{x}))$ and $\hat{\eta}_{\min}(r(\mathbf{x}))$. This selection limits the number of figures shown, but the qualitative conclusions remain the same as in other cases (e.g., using baseline parameters from different folds or shifting between arbitrary pairs of $\hat{\eta}(r(\mathbf{x}))$).

A.6 FITTING PERFORMANCE

Tables 2–4 report the performance of power-law fitting and baseline shifting, evaluated with MAE, RMSE, and HL across different dimensions. Metric values are averaged over three folds in the estimation of the shifting parameter. The baseline parameters are taken from the first fold of the cross-validation during power-law fitting. Using baseline parameters from other folds yields similar trends and is therefore omitted here.

