# OpenReview forum: "Effect of Local Oscillations on the Scaling Laws of Deep Neural Networks"
_ICLR.cc/2026/Conference — ICLR 2026 Conference Withdrawn Submission_

### Official Review · Reviewer_Hg9z · 2025-10-27

**Soundness:** 3
**Presentation:** 3
**Contribution:** 2
**Rating:** 4
**Confidence:** 3

**Summary:**

In deep learning, scaling laws predict e.g. test loss as a function of data and network properties. Current estimation methods do not take into account whether the data shows oscillatory behavior (as e.g. in the case of the parity function, the task of predicting whether the number of ones in a binary input vector is odd or even). Through a combination of theory and experiments on feedforward multilayer perceptrons, this article shows that this local oscillatory behavior - termed LOC - "changes both exponent and prefactor of the scaling law".

Caveat regarding my review: I'm not a theorist and am not familiar enough with the math to provide an informed assessment of the proofs. I'm familiar with the overall area and the empirical side.

**Strengths:**

1. The article is very well written overall and a pleasure to read
2. Clearly structured around three research questions (Q1-- Q3), which are then answered one-by-one, makes it easy to follow
3. scaling laws are an important area of research, and characterising the conditions under which scaling can be predicted well is valuable & of interest to the research community

**Weaknesses:**

1. Unclear relevance: I'm not convinced yet that the insights are of any practical relevance beyond "ivory tower theory". To be clear, I would love to be convinced otherwise, but it's up to the authors to convince a reader why they should care about the experiments and theory. They define LOC but do not show which real-world data, if any, (e.g. across vision, language, audio, health, finance, ... data) exhibits significant local oscillatory behavior.

2. Choice of architecture for empirical experiments is outdated by 10+ years. While I understand that multilayer perceptrons are easier to characterise theoretically, on the empirical side it's unclear why this architecture was chosen as opposed to more up-to-date architectures such as transformers. It is known from both theory and experiments that the scaling behavior of MLPs can be quite different from that of a transformer, and transformers are used by the vast majority of researchers/engineers when it comes to real-world problem solving with deep learning techniques currently. It would be valuable to show that the theoretical predictions and empirical characterizations derived from these settings transfers to a setting that's of practical relevance and uses a common choice of architecture and dataset. Furthermore, it's unclear why feedforward perceptrons were chosen in the first place, since Han et al. 2025 specifically find that "transformers [are] theoretically superior to FFNNs in learning parity function".

3. Toy task (binary classification) also limits real-world relevance.

4. Are parity functions relevant for any real-world use cases and for neural networks in particular? I understand that parity functions are a good choice for the theory to characterize LOC but I'm not sure whether they matter in practice (and would love to be convinced). If neural networks are known to have a low frequency bias (as the authors write themselves), why would we choose this function family for learning the parity function in the first place? According to the no free lunch theorem, no single algorithm is universally good, thus if the algorithm's inductive biases are a poor fit for the properties of the data, shouldn't we then use a different algorithm?

**Questions:**

MISC suggestions for improvement:
- Figure clarity: many readers skim a paper before reading it; it's helpful if a paper's core ideas can be understood from looking at the figures. This is currently not the case. Figure 1, for example, could easily be turned into a more self-explanatory version by not just annotating the axes with symbols (N, l) but also a natural-language description. Example: "Dataset size (log_n)" is a lot more easy to understand as an axis annotation than "log n" since it assumes no prior knowledge on the reader's side. The same holds for other annotations, legends and figures.
- the same holds for Table 1: it would be helpful to explain the abbreviations in the caption and also state what the reader is supposed to take away from the table. For example, I personally know what RMSE is, but not what e.g. HL means in this context.

---

### Official Review · Reviewer_nWfU · 2025-10-27

**Soundness:** 3
**Presentation:** 2
**Contribution:** 1
**Rating:** 2
**Confidence:** 5

**Summary:**

In this paper the authors investigate the utility of Holder-smoothness based generalization bounds by constructing functions with similar Holder-smoothness properties but different local oscillation structure. The local oscillation structure is defined using the norm of the gradient of the function with respect to inputs. The authors show that the family of $\cos(\omega ||x_{1}||)$ has constant Holder complexity but different oscillation scales.

**Strengths:**

I believe the theoretical analysis is correct. The overall point that Holder class is insufficient to give good understanding of generalization behavior is also sound.

**Weaknesses:**

Overall the paper claims to be an exploration of scaling laws; however I believe the setup is not appropriate to investigate this area.

The notion of "scaling laws" is generally poorly defined but to me refers to the "empirical power law scaling" behavior of many extensive quantities (most notably, the loss) in settings where models are trained efficiently (indeed in many cases, compute-optimally under various constraints). The exponents that arise in these studies are highly non-trivial and depend mostly on data, but sometimes on architecture and optimization rules.

The experiments in this paper are on a very simple, very low dimensional function. This is not the appropriate regime to study the empirical power laws that occur in large models, which arise due to a combination of features at different scales, as well as feature learning over many steps. Even though the paper does attempt to understand some scaling relations (e.g. sample complexity), it does not focus on issues which are relevant to the most practically relevant regimes of "empirical power law scaling".

The setting of the work is most similar to the work on single/multi-index models, where generalization bounds are written in terms of the information exponent e.g. [1, 2]. It is also reminiscent to past work analyzing XOR-style functions in a similar setting [3]. All these works managed to obtain a much more detailed analysis of the relationship between function geometry and generalization and these (and similar approaches) are now the standard in the area. This current work only shows that Holder class bounds are loose and misleading. The literature as a whole has moved beyond such bounds to ones that are more detailed and geometric as evidenced by [1, 2, 3] (and of course many other related works).

[1] https://proceedings.neurips.cc/paper_files/paper/2023/hash/02763667a5761ff92bb15d8751bcd223-Abstract-Conference.html
[2] https://arxiv.org/abs/2504.19983
[3] https://proceedings.neurips.cc/paper_files/paper/2022/hash/a224ff18cc99a71751aa2b79118604da-Abstract-Conference.html

**Questions:**

What is the connection between the theoretical ideas in this paper, and the literature on single index/multi index models? In particular there are many works which try to analyze generalization in terms of the information exponent, e.g. [1] and [2].

$\eta$ is often used as the learning rate in optimization algorithms; it may be better to choose a different variable name for the oscillatory measures. Additionally, it should be made clear that the gradient in this definition is with respect to inputs; the most common gradient in machine learning is with respect to parameters so readers may fill in the blanks incorrectly.

[1] https://proceedings.neurips.cc/paper_files/paper/2023/hash/02763667a5761ff92bb15d8751bcd223-Abstract-Conference.html
[2] https://arxiv.org/abs/2504.19983

---

### Official Review · Reviewer_Ymm6 · 2025-11-04

**Soundness:** 3
**Presentation:** 3
**Contribution:** 2
**Rating:** 4
**Confidence:** 3

**Summary:**

The authors study how the local oscillatory complexity (LOC) of a target function can significantly change the scaling behavior of a model. LOC describes how quickly the target function's output changes locally.

Specifically, they show that when the target function's LOC is high (closer to a parity function), additional data is unlikely to help the test loss. They show this empirically on DNNs.

**Strengths:**

- The paper is clearly written and presented
- The authors claims are backed by experiments on DNNs
- The LOC metric is a nice measure related to other measures of target function complexity
- This crux of this paper boils down to showing that for a given model capacity, overly complex target functions can negatively affect the scaling function. This aligns with what is seen in practice.

**Weaknesses:**

- It's not very clear whether LOC describes anything more than target functions being too complex, which do have measures in place already such as intrinsic dimension, Lipschitz constant already. It is not surprising that functions that are complex possibly need larger model sizes.
- Given this interpretation of the paper, the draft would be significantly stronger if the effect of model size on the conclusions was studied.

**Questions:**

- How does LOC connect to real world datasets such as text and images and categorical data?
  - Eg "In fact, if a model were trained on a
low-oscillation dataset and tested on a high-oscillation scenario, our results indicate its performance
could be much worse than predicted by simply extrapolating the training scaling law" How would this manifest in an example real world dataset?
- In what way does changing LOC instead of any other measure of target function complexity explain this degradation in scaling law better? I would be willing to change scores if discussed and highlighted sufficiently.

---

### Official Review · Reviewer_GwR9 · 2025-11-04

**Soundness:** 2
**Presentation:** 3
**Contribution:** 1
**Rating:** 2
**Confidence:** 4

**Summary:**

The paper defines a measure of Local Oscillatory Complexity (LOC) and considers a class of functions parametrized by a frequency parameter that leads to a range of different LOC. It is then observed empirically that changing the frequency parameter does has a more qualitative impact on the scaling laws, not only changing the prefactor but possibly changing the exponent.

**Strengths:**

The paper is quite clear and well written and it studies very simple tasks but observes some non-obvious behavior.

**Weaknesses:**

The "surprising behavior" observed in this paper is only surprising
from a shallow interpretation of how Hölder/Sobolev continuity
affects the scaling law. the bounds of (Schmidt-Hieber, 2020) are
only upper bounds, they never predict that the log-log plot of the
test error will be close to linear. To more precisely predict the
learning curve, people have used Random Matrix Theory (RMT), which
roughly speaking predicts that for a given number of datapoints $N$
and number of parameters $M$ the function will capture the signal
along the first $\min\{N,M\}$ principal components (PCs) of the data
(see e.g.  https://arxiv.org/abs/2006.09796).

We can therefore expect a nice linear log-log plot if the signal is
roughly evenly spread-out over the PCs (or rather evenly decaying), however the function you consider
has almost all of its signal concentrated in a small range of frequencies
(close to your $\omega$ parameter), which leads to a sigmoid-type
of loss behavior. This is exactly what you observe, and it fits very
well within the existing theory. Changing $\omega$ translates the sigmoid on the $N$-axis

The functions considered are very artificial and synthetic, and so
I expect these sigmoid type of loss curves to be very rare on practical
data. If you consider unusual tasks, you will observe unusual loss
behavior, as expected.

**Questions:**

- The LOC is not merely “related to” Sobolev norms, it is almost exactly a $L_1$-Sobolev norm. I don't see how the LOC is better adapted to this question, because the Sobolev norm would exhibit essentially the exact same behavior on the cosine functions considered. If you want to introduce a new definition and even a whole new fancy name for it, then you better have a strong justification for why it is better than the already existing concepts, and I think that it is clear that it is not necessary. It also makes everything more confusing, because if the abstract talked about Sobolev norm then it would be obvious what the paper is doing right away.

- I feel like the network size $M$ should also be increased as $N$ grows in the experiments, otherwise some of your conclusion could instead be an effect of network size.

---

### Note · Authors · 2025-11-20

I have read and agree with the venue's withdrawal policy on behalf of myself and my co-authors.